# Preparation of Gelatin-Quaternary Ammonium Salt Coating on Titanium Surface for Antibacterial/Osteogenic Properties

**DOI:** 10.3390/molecules28124570

**Published:** 2023-06-06

**Authors:** Hongyang Song, Lei Xing, Jinjian Wei, Xue Wang, Yaozhen Yang, Pengbo Zhe, Mingming Luan, Jing Xu

**Affiliations:** 1Shandong Provincial Key Laboratory of Molecular Engineering, School of Chemistry and Chemical Engineering, Qilu University of Technology (Shandong Academy of Sciences), Jinan 250353, China; hy2495756023@163.com (H.S.); xl2713698788@163.com (L.X.); wxue202111@163.com (X.W.); 15266726773@163.com (Y.Y.); 18991783750@163.com (P.Z.); 2College of Chemistry, Chemical Engineering and Materials Science, Shandong Normal University, Jinan 250100, China; jinjian.wei@sdnu.edu.cn

**Keywords:** Ti, osseointegration, antibacterial, electrostatic self-assembly technology

## Abstract

Titanium (Ti) and its alloys are widely used in medical treatment, engineering, and other fields because of their excellent properties including biological activity, an elastic modulus similar to that of human bones, and corrosion resistance. However, there are still many defects in the surface properties of Ti in practical applications. For example, the biocompatibility of Ti with bone tissue can be greatly reduced in implants due to a lack of osseointegration as well as antibacterial properties, which may lead to osseointegration failure. To address these problems and to take advantage of the amphoteric polyelectrolyte properties of gelatin, a thin layer of gelatin was prepared by electrostatic self-assembly technology. Diepoxide quaternary ammonium salt (DEQAS) and maleopimaric acid quaternary ammonium salt (MPA−N^+^) were then synthesized and grafted onto the thin layer. The cell adhesion and migration experiments demonstrated that the coating has excellent biocompatibility, and those grafted with MPA−N^+^ promoted cell migration. The bacteriostatic experiment showed that the mixed grafting with two ammonium salts had excellent bacteriostatic performance against *Escherichia coli* and *Staphylococcus aureus*, with bacteriostasis rates of 98.1 ± 1.0% and 99.2 ± 0.5%, respectively.

## 1. Introduction

In the 1950s, Ti was used as an animal implant material, and thereafter caught the attention of researchers. In the 1960s, Ti was successfully implanted into human bodies in clinical surgical trials in the United States and Britain. Ti has an elastic modulus similar to that of human bones, as well as higher fatigue strength and toughness compared to other metal materials. These excellent properties make it the best choice for replacing and repairing human bone tissues as artificial bones, joints, and dental implants [1,2,3,4,5,6,7,8,9,10]. However, Ti has defects in its surface property, such as the tendency to form oxide coatings, which cause biological inertness and results in a lack of “active repair” function. This can lead to the inability of Ti materials to firmly bond with surrounding tissues after implantation, causing the loss of osseointegration ability and resulting in implant loosening or detachment. In addition, the chemical instability and deformation of the protective oxide layer usually leads to poor osseointegration, and its deformed surface releases metal ions to form Lewis acid, lowering the pH value in the implantation environment. Bacterial infection and weak osseointegration will result in implant surgical failure, which makes Ti-based implants unsuitable for prolonged use [9]. Implant infection may cause suppuration, revision surgery, or even the need to be re-implanted after removal. This may not only necessitate extremely expensive medical costs but also aggravate patients’ pain [11]. Previous research findings [12,13,14] have indicated several possible modifications to overcome these defects. In consideration of biocompatibility, antibacterial materials can be introduced, which not only improve the shortcomings of the monolayer membrane of gelatin (such as relatively poor mechanical properties, strong moisture absorption and fast degradation rate), but also subdue the problem of bacteria breeding on the surface. To extend the functional properties of these biodegradable films and to make them more durable, related technologies are expected to advance the field of bone integration. Therefore, bacterial inhibition/elimination and osseointegration promotion are desirable properties for Ti-based orthopedic implant materials [15].

Collagen, with a triple-helix structure composed of α-chain peptides, is one of the important rich substances in the extracellular matrix of vertebrates and plays an important role in the integrity and mechanical properties of biological tissues [16,17,18,19,20]. The network structure of collagen in cartilage is the foundation of the joint structure, which provides the joint with a certain degree of flexibility and stability. Mature bone matrix contains a large number of organic substances such as collagen fibers and mucin, which provide elasticity and toughness to bones.

For decades, type I collagen has been the preferred option for bionic design in tissue engineering [21,22]. It is composed of three polypeptide α chains with intermolecular Gly-X-Y repeat units, where Gly is glycine, X is often proline (Pro) and Y is always hydroxyproline (Hyp). Gelatin is composed of the triple-helix structures of type I collagen that completely open and gradually decompose into triple-helix configurations, namely: single-stranded (α Chain), double α Chain (β Chain), and triple α Chain (γ Chain) [16,23].

Gelatin is characterized by natural richness, a relatively low price, non-toxicity, and good biocompatibility in a physiological environment. It can promote cell adhesion and proliferation, induce cell differentiation, and serve as a scaffold for cell growth. Gelatin can be manufactured in various forms, including films, micro- or nano-particles, dense or porous hydrogels, and fiber electronic spinning technology.

The molecular chain of gelatin comprises functional groups such as -CH_2_, -NH_2_, -COOH and -OH. According to Krajewski et al. [24], the charge on the material’s surface is one of the main physical factors affecting the biological binding of the tissue around the implant. Both the amount and distribution of charges will affect the adhesion strength of the cells around the implant and the shape of the cells when they come into contact with the surface of the material. Our research group [25] found that the exposure of these groups in the gelatin chain can be controlled by regulating the conformation of gelatin, therefore achieving the coordinated regulation of surface charge properties. Using self-assembled monolayer technology [26,27,28,29,30], a strategy for assembling a gelatin monolayer on a Ti surface by regulating the conformation of gelatin was devised to achieve the regulation of its surface properties [31,32,33,34,35,36,37,38], and the gelatin monolayer was prepared as shown in Figure 1. However, the gelatin monolayer exhibits relatively poor mechanical properties, strong hygroscopicity, and a fast degradation rate. The probability of expansion or dissolution increases when it comes into contact with a surface that has a high moisture content. In addition, the surface is easy to breed bacteria on, which limits its application in bioengineering, industrial production, and other fields. As a result, gelatin monolayers need to be enriched with antibacterial substances/natural antioxidants in order to increase the functional capabilities and durability of these biodegradable films [39].

In the present study, gelatin monolayers were assembled on the surface of Ti using the electrostatic method. Quaternary ammonium bis epoxide salt (DEQAS) and quaternary ammonium Balearic acid salt (MPA−N^+^) were synthesized and then grafted onto the collagen monolayer. Cell migration and the antibacterial properties of the resulting Ti implant were studied to evaluate its potential used in the field of osseointegration.

## 2. Results and Discussion

### 2.1. Calculation of Molar Grafting Rate

The molar grafting rate of quaternary ammonium salt was calculated as per the formula in Section 3.5. Table 1 shows that the grafting rate of quaternary ammonium salt was maximum when the concentrations of sodium dodecyl sulfate (SDS) and sodium tetradecyl sulfonate (STSo) were 8.32 mmol/L and 7.96 mmol/L, respectively. Our previous IR study indicated that the epoxy groups reacted with the amine groups [40].

### 2.2. Determination of Wettability of Coating Surface

The optical contact angle (WCA) grafted with quaternary ammonium salt was measured with an optical contact angle measuring instrument. As shown in Figure 2, all WCAs of DEQAS and MPA−N^+^ grafted with different systems of gelatin monolayers were larger than those of the grafted EPTAC, with the WCA of the quaternary ammonium salt Gelatin (G) −grafting system (SDS dosage was 6% of the mass of gelatin, SDS_6%_) being slightly higher than that of the G−SDScac (SDS_cac_ is the critical aggregation concentration of SDS); STSo_6%_ grafting system higher than that of the G- STSocac (STSocac is the critical aggregation concentration of SDS). The WCA was changed from 9.0 ± 0.9° to 68.1 ± 0.9° and 64.1 ± 0.9° when the G−SDS dosage was 6% of the mass of gelatin; SDS_6%_ was grafted with DEQAS and MPA−N^+^, respectively.

The above results showed that the WCA of collagen monolayer grafted with quaternary ammonium salt was larger than that of the gelatin monolayer, but it still exhibited hydrophilicity, which was beneficial to osseointegration.

### 2.3. Analysis of Coating Surface Morphology

The samples prepared above were characterized for the coating surface morphology using optical microscopy (OM), and the smoothness of the coating surface of the gelatin monolayer was studied after grafting with different quaternary ammonium salts. Because the molar grafting rates of SDS_6%_ and STSO_6%_ quaternary ammonium salt grafting systems were the largest, the surfactant systems selected in this test were SDS_6%_ and STSO_6%_. Figure 3 shows the results of grafting various quaternary ammonium salts after preparing thin gelatin layers. According to the OM images, the coating surface after grafting with MPA−N^+^ or DEQAS, as well as grafting with mixed MPA−N^+^ and DEQAS, were all dense and smooth, indicating that MPA−N^+^ and DEQAS were successfully grafted. In addition, the surface of the grafted coating was very uniform, which would be more suitable for osseointegration.

### 2.4. Determination of Surface Roughness of the Coating

The surface Ra of the coating prepared above was determined by AFM. As shown in Figure 4, the gelatin monolayer (Figure 4a,f) was composed of tightly packed spherical nanoparticles. The average size of the nanoparticles in a 6% wt SDS system was about 60 nm, and Ra was 5.76 nm. The average size of the nanoparticles in a 6% wt STSo system was about 30 nm, and Ra was 5.02 nm.

The surfaces of various quaternary ammonium salts grafted with G−SDS_6%_ and G−STSo_6%_ were relatively flat and uniform. The Ra of the coated surface rose from 5.67 nm to 14.78 nm when G−SDS_6%_ was grafted with MPA−N^+^, while the Ra of the combined grafting with DEQAS and MPA-N^+^ increased to 18.60 nm, both of which were higher than that of the grafting with any quaternary ammonium salt. The Ra of the coated surface rose from 5.02 nm to 15.48 nm when G−STSo_6%_ was grafted with MPA−N^+^, while the Ra of the combined grafting with DEQAS and MPA−N^+^ increased to 19.40 nm. Ra in the G−SDS_6%_ and G−STSo_6%_ systems was changed similarly after grafting with different quaternary ammonium salts, showing an increasing trend with the increase in molecular weight of quaternary ammonium salts (Figure 5 and Figure 6). The coated surface grafted with EPTAC has the lowest Ra. This is because more primary amino groups could carry out ring-opening reactions with epoxy groups under alkaline conditions, resulting in stronger interactions between groups, and consequently the molecules could be closely distributed on the surface of the gelatin monolayer due to the small molecular weight of quaternary ammonium salts. The reason why the Ra of the coating surface grafted with DEQAS and MPA−N^+^ was the largest was that the molecular weight of these two quaternary ammonium salts was larger than that of EPTAC. The increase in Ra could stimulate the formation of bone tissue around the implant to a certain extent, thereby shortening the osseointegration process.

### 2.5. Determination of Cell Adhesion on Coating Surface

Although gelatin has excellent biocompatibility, quaternary ammonium salts and surfactants can cause cell irritation. The crystal violet staining cell counting method was used for determining cell adhesion. In the initial process of osseointegration, the adhesion and proliferation of cells on the surface of implanted materials play an important role. Therefore, detecting the capability of cell adhesion on the surface of implanted materials is important in determining whether or not the material can exert biological activity. The results showed that all samples exhibited cell adhesion properties, with similar patterns observed in the SDS_6%_ and STSo_6%_ systems. As shown in Figure 7, the numerical order of surface cell adhesion is: Control > G(STSo_6%_)−MPA-N^+^ > Blank > Gelatin > G(STSo_6%_)−DEQAS/MPA-N^+^ > G(STSo_6%_)−DEQAS > G(STSo_6%_) > G(STSo_6%_)−EPTAC. The cell adhesion performance of grafting with MPA−N^+^ was the best, followed by grafting with DEQAS and MPA−N^+^. The cell adhesion performance of DEQAS grafting was better than EPTAC grafting. These results indicate that the gelatin coatings grafted with quaternary ammonium salt are not toxic to cells and are therefore suitable for use in osseointegration materials.

### 2.6. Determination of Cell Migration on Coating Surface

The scratch test was used to determine the effect of the various coating samples on cell migration. Using the STSo_6%_ system as an example, the results indicated that, compared with the cells grown on the culture plate, the cells on different coatings had varying degrees of migration in the scratch area (Figure 8). The results of cell mobility were g > b > a > c > h > f > d > e, indicating that trace surfactants had little effect on cell mobility. The cell compatibility of the coating grafted with quaternary ammonium salts was greater than that of the DEQAS grafted coating but lower than that of the MPA−N^+^ grafted coating.

### 2.7. Determination of Antibacterial Property of Coating Surface

Antibacterial research mainly focuses on the inhibitory effect of materials on the bacteria around them. Using the STSo_6%_ system as an example, the antibacterial performance of each coating was qualitatively analyzed using the plate colony counting method in order to evaluate the antibacterial activity of coatings. Figure 9 and Figure 10 demonstrate that metal Ti alone, which had no antibacterial properties, produced a large number of bacteria after 24 h of culture in *S. aureus* and *E. coli*. As compared to metal Ti, the coating grafted with quaternary ammonium salts had excellent antibacterial properties, and the antibacterial rates after being grafted with DEQAS on *E. coli* and *S. aureus* were 91.8 ± 2.0% and 97.6 ± 2.0%, respectively. Because DEQAS has more epoxy groups and a higher grafting rate, the antibacterial properties of the coating grafted with DEQAS are obviously better than those grafted with EPTAC. The antibacterial rates of grafting with MPA−N^+^ against *E. coli* and *S. aureus* were 90.9 ± 2.0% and 97.9 ± 2.0%, respectively. The antibacterial property of grafting MPA−N^+^ was clearly better than that of grafting DEQAS. The antibacterial rates of DEQAS and MPA−N^+^ against *E. coli* and *S. aureus* were 98.1 ± 1.0% and 99.2 ± 0.5%, respectively. The antibacterial properties of the mixed grafting with DEQAS and MPA−N^+^ were better than those of the grafting with one quaternary ammonium salt due to the anti-inflammatory advantages of the raw materials for synthesizing MPA−N^+^.

## 3. Materials and Methods

### 3.1. Materials

Gelatin (type A, extracted from pig skin, MW ≈ 50,000 g mol^−1^) was supplied by China Tianjin Traditional Chinese Medicine Group Corporation without further purification at the time of use. Crystal violet (AR), peptone (BR), (2,3-Epoxypropyl) trimethylammonium chloride (EPTAC, AR), 2-(N-morpholinyl) ethanesulfonic acid buffer solution (MES, AR), sodium tetradecyl sulfate (STSo, AR), ethidium bromide (AR), N-hydroxysuccinimide (NHS, AR), N,N,N’,N’-Tetramethylethylenediamine (TMEDM, AR), Epichlorohydrin (EC), 1-ethyl-(3-dimethylaminopropyl) carbodiimide (EDC, AR), Abietic acid (AR) were provided by McLin Biochemical Technology Co., Ltd. Sodium dodecyl sulfate (SDS, AR), sodium carbonate (AR), sodium bicarbonate (AR) and potassium sulphate (AR) were provided by Aladdin Biochemical Technology Co., Ltd. (Shanghai, China), and anhydrous ethanol was provided by Tianjin Fuyu Fine Chemical Co., Ltd. (Tianjin, China) Ti (specification: 1 cm × 1 cm, thickness 1 mm) was provided by Guangdong Als Metal Technology Co., Ltd. (Guangzhou, China).

### 3.2. Preparation of Collagen Peptide Solution

Synthesis of DEQAS [40]: A 250.0 mL three-port flask was taken and distilled water (22.0 mL), K_2_SO_4_ (0.2 g), methanol (17.0 mL) and epichlorohydrin (EC, 9.5 g) were added to it in turn. Then, the mixture in the three-port flask was heated to 50 ± 1 °C and stirred at constant temperature for 0.5 h. During this time, tetramethylethylenediamine (TMEDM) was added to the flask at a rate of 12 drops/min. The reaction was stopped after stirring for 1.5 h. The solution was poured into a 250 mL round-bottom flask and distilled under reduced pressure to obtain a yellowish liquid (DEQAS) (Figure 11).

Synthesis of MPA−N^+^: The rosin acid-derived quaternary ammonium maleic acid cation (MPA−N^+^) was prepared using a method detailed in the literature [41], and with the reaction route being shown in Figure 12. The Abietic acid (100.0 g, 0.28 mol) was heated to 180 °C and refluxed for 3 h under nitrogen atmosphere, and the heated rosin acid was cooled to 120 °C, after which maleic anhydride (27.5 g, 0.28 mol) and acetic acid (400.0 mL) were added to the above reaction system. The reaction was refluxed at 120 °C for 12 h. Then, the reaction was cooled to room temperature and left for another 2 h. The crude maleic pine acid was recrystallized twice in acetic acid to obtain pure maleic pine acid (MPA, 91.0 g, 97% purity, 79% yield).

Maleic acid (MPA, 10.0 g, 0.025 mol) was dissolved in ethanol (250.0 mL). Then, N,N-dimethylethylenediamine (2.8 mL, 0.025 mol) was added and refluxed at 85 °C for 5 h. The solution was cooled to room temperature, crystallized, filtered and dried to obtain the product MPA−N (9.5 g, purity: 98%, yield: 79%).

MPA−N (1.0 g, 0.0021 mol) and bromoethane (3.1 mL, 0.043 mol) were dissolved in redistilled tetrahydrofuran (THF, 30.0 mL) and reacted at 40 °C for 48 h. Then, the product MPA−N^+^ (0.94 g, purity: 90%, yield: 70%) was crystallized, filtered and dried.

#### Synthesis and Characterization of DEQAS and MPA−N^+^

Refer to our previous literature for DEQAS synthesis [40]. Figure 13 shows the ^1^H NMR spectrum derived from the structural characterization of MPA, MPA−N, and MPA−N^+^. ^1^H NMR(DMSO, 400 MHz), δ(TMS, ppm), 12.08(s, ^1^H), 5.52(s, 1H), 3.23(s, 1H), 2.91(s, 1H), 2.33(s, 1H). Figure 13b ^1^H NMR (CDCl_3_, 400 MHz), δ(TMS, ppm), 12.16(s, 1H), 5.42(s, 1H), 3.71(t, 2H), 3.41(s, 1H) and 3.19(s, 1H). Figure 13c ^1^H NMR (DMSO, 400 MHz), δ(TMS, ppm), 12.16(s, 1H), 5.38(s, 1H), 3.63(t, 2H), 3.40(t, 2H) and 3.17(s, 2H). 1.68(m, 11H), 1.45(s, 4H), 1.23(t, 3H), 1.04(s, 3H), 0.90(t, 6H), 0.53(s, 3H). The attribution of each peak was determined using information such as various chemical shifts of protons and different chemical environments.

### 3.3. Preparation of Collagen Peptide−Quaternary Ammonium Salt Coatings

Preparation of collagen peptide monolayers: (1) Preparation of seven 50 mL of 4% wt collagen peptide solution. The collagen peptide and distilled water were added to a single-bottom flask and left to swell for 30 min at 25 °C. The solution was heated and stirred at 50 °C for 30 min, and then different masses of SDS were added to the three collagen peptide solutions to make the concentrations of SDS in the collagen peptide solutions 3.50 (critical aggregation concentration CAC) mmol/L (SDS_cac_), 7.50 (critical micelle concentration CMC) mmol/L (SDS_cmc_), 8.32 (6% wt) mmol/L (SDS_6%_), respectively; finally, different masses of STSo were added to the other three collagen peptide solutions so that the concentrations of STSo in the collagen peptide solutions were 2.50 (CAC) mmol/L (STSo_cac_), 7.00 (CMC) mmol/L(STSo_cmc_), and 7.96 (6% wt) mmol/L(STSo_6%_). Finally, all the above solutions were heated and stirred at 50 °C for 6 h at constant temperature. (2) The titanium sheets were manually polished with metallographic sandpaper in different orders of 800, 1500, 3000, 5000, and 7000 mesh, and ultrasonically cleaned with distilled water, anhydrous ethanol, and acetone for 15 min each, and then the cleaned titanium sheets were blown dry with high-purity nitrogen. The 30% H_2_O_2_ and 98% H_2_SO_4_ solution were mixed in equal volume ratios to configure the mixed acid solution. After the mixed acid solution was cooled to room temperature, the dried titanium flakes were soaked in the mixed acid solution for 1 h, washed with distilled water to neutral, and then the cleaned titanium flakes were blown dry with high-purity nitrogen. At room temperature, the acid-treated titanium flakes were placed in 10% polyethylenimine (PEI) solution for 30 min, washed with distilled water 5 times, and finally the cleaned titanium flakes were blown dry with high-purity nitrogen and dried in a constant temperature drying oven at 60 °C for 12 h. The titanium flakes treated with PEI solution were put into the reaction flask, and then the solutions prepared in the reaction flask were added into the reaction flask separately and left to be deposited at 50 °C for 10 min; the flasks were clamped out with hemostats and then put into distilled water and lifted up and down uniformly for 20 times. Finally, the deposited titanium flakes were blown dry with high-purity nitrogen. After Ti flakes were denoted by G, i.e., G−SDS_cac_, G−SDC_cmc_, G−SDS_6%_, G−STSo_aca_, G−STSo_CMC_, G−STSo_6%_, 7 samples were prepared, including Ti−G, Ti−G−SDS_cac_, Ti−G−SDC_cmc_, Ti−G−SDS_6%_, Ti−G−STSo_aca_, Ti−G−STSo_CMC_, and Ti−G−STSo_6%_. Blank is Ti−G.

Preparation of monoepoxy quaternary ammonium salt (EPTAC) grafted collagen peptide coatings: Firstly, Na_2_CO_3_/NaHCO_3_ buffer with pH 9.6 was configured, 5 mL of buffer solution was added to the reaction flask, EPTAC (29 mg) was added, and the above reaction flask was placed in an ultrasonic cleaner for 10 min so that EPTAC could be completely dissolved in the buffer solution, and to a final concentration of 5.8 mg/mL. Then, the prepared sample was placed in the reaction flask in a constant-temperature water bath at 50 °C for 12 h. The weakly bound or unbound quaternary ammonium salts were removed by lifting up and down uniformly in distilled water 10 times, and then blown dry with high-purity nitrogen and stored in nitrogen gas. The resulting coating was labeled as collagen peptide−EPTAC.

Preparation of DEQAS grafted collagen peptide coatings: Firstly, Na_2_CO_3_/NaHCO_3_ buffer with pH 9.6 was configured, 5 mL of buffer was added to the reaction flask, 10 drops of DEQAS (37 mg) were added with a 1 mL syringe, and the above reaction flask was sonicated for 10 min so that DEQAS could be completely dissolved in the buffer; then, the prepared samples were placed in the reaction flask and reacted in a constant-temperature water bath at 50 °C for 12 h. The samples were then placed in the reaction flask, and the reaction was carried out in a constant-temperature water bath at 50 °C for 12 h. The samples were lifted up and down uniformly in distilled water 10 times to remove the weakly bound or unbound quaternary ammonium salts, blown dry with high-purity nitrogen and stored in nitrogen. The resulting coating was labeled as collagen peptide-DEQAS.

Preparation of MPA−N^+^ grafted collagen peptide coatings: To the reaction flask were added 5 mL of 2-(N-morpholinyl)ethanesulfonic acid (MES) buffer solution (0.1 mol/L, pH = 5.5), EDC (1-(3-dimethylaminopropyl)-3-ethylcarbodiimide hydrochloride), NHS (N-hydroxythiosuccinimide), and MPA−N^+^ 42 mg (where n(MPA−N^+^):n(EDC) = 1:427.35; n(MPA−N^+^):n(NHS) = 1:854.70). The reaction flask was then sonicated for 10 min so that MPA−N^+^ could be completely dissolved in the buffer solution (the concentration of MPA−N^+^ was 0.0144 mol/L), after which the prepared samples were placed in the reaction flask and reacted in a constant-temperature water bath at 50 °C for 12 h. The samples were lifted up and down in distilled water 10 times to remove the weakly bound or unbound quaternary ammonium salts, blown dry with high-purity nitrogen and then stored in nitrogen gas. The resulting coating was labeled as gelatin−MPA−N^+^.

Preparation of DEQAS and MPA−N^+^ hybrid grafted collagen peptide coatings: Firstly, Na_2_CO_3_/NaHCO_3_ buffer solution with pH 9.6 was configured, 5 mL of buffer solution and DEQAS (21.3 mg) was added to the reaction flask, and the above reaction flask was placed in an ultrasonic cleaner for 10 min to completely dissolve DEQAS in the buffer solution (the concentration of DEQAS was 0.0142 mol/L); then the prepared sample was placed in the reaction flask and reacted in a constant-temperature water bath at 50 °C for 12 h. The weakly bound or unbound quaternary ammonium salts were removed by lifting up and down in distilled water for 10 times, and then thoroughly blown dry with high-purity nitrogen and stored in nitrogen, and the resulting coating was labeled as collagen peptide−DEQAS, NHS (N-hydroxythiosuccinimide) and MPA-N^+^ (20.30 mg) (where n(MPA-N^+^):n(EDC) = 1:427.35; n(MPA-N^+^):n(NHS) = 1:854.70). The reaction flask was sonicated for 10 min to completely dissolve MPA−N^+^ in the buffer solution (the concentration of MPA−N^+^ was 0.00173 mol/L); then the sample gelatin−DEQAS was placed in the above reaction flask and reacted under the condition of a constant-temperature water bath at 50 °C for 12 h. The weakly bound or unbound quaternary ammonium salts were removed by lifting up and down uniformly in distilled water 10 times, and the was thoroughly blown dry with high-purity nitrogen and stored in nitrogen gas. The resulting coating was labeled as collagen peptide−DEQAS/(MPA-N^+^).

### 3.4. Nuclear Magnetic Resonance Hydrogen Spectroscopy (1H NMR) Characterization

The synthesized products were structurally characterized using an NMR instrument (model: AVANCE II 400 MHz, Bruker, Billerica, MA, USA) with tetramethylsilane (TMS) as an internal standard; the volume of deuterium reagent used was 0.6 mL, and the number of scans was 64. The ^1^H NMR spectra obtained were data processed, analyzed and discussed using Mestranova software to confirm the structure of the product.

### 3.5. Molar Grafting Rate Calculation

In order to simulate a Ti sheet, pure Ti powder was pressed into a 75 mm diameter disk. A thin layer of Ti was deposited on a quartz disk (diameter: 25 mm) by reactive magnetron sputtering using a radio-frequency magnetron sputtering system (CFS-4ES-231). The magnetron sputtering chamber was evacuated to a pressure and kept at 6.7 × 10 ^−1^ Pa with argon, resulting in a Ti layer. The Ti layers were cleaned with sodium dodecyl sulfate and UV-ozone cleaner before the QCM measurements. In QCM, ΔF depends on the adsorbed mass following Sauerbrey’s equation:$$\Delta F=-\frac{2F_{0}^{2}}{\sqrt{\rho_{q}\mu_{q}}}\frac{\Delta m}{A}$$
where, F_0_ is the fundamental frequency of the crystal (27 × 10^6^ Hz), A is the electrode area (0.049 cm^2^), ρ_q_ is the quartz density (2.65 g/cm^3^), and μ_q_ is the shear modulus of quartz (2.95 × 10^11^ dyn/cm).

The masses of the samples prepared above were weighed accurately using a quartz crystal microbalance (model: QSense Initiator, Gothenburg, Sweden), and the changes in the masses of the quaternary ammonium salts before and after grafting were recorded, repeated three times, and averaged. The molar grafting rate of quaternary ammonium salts was calculated using the following equation [42].
$$\mathrm{Moore}\;\mathrm{grafting}\;\mathrm{rate}=\frac{W_{D}-W_{0}}{{M_{W}\times W}_{0}}\times 100\%$$
where W_D_ is the mass after grafting the quaternary ammonium salt, W_0_ is the mass before grafting the quaternary ammonium salt, and M_W_ is the molecular weight of the quaternary ammonium salt.

### 3.6. Optical Contact Angle (WCA) Measurement of the Coating Surface

At room temperature, WCAs of the above prepared Ti specimens were measured using an optical contact angle measuring instrument (DSA-100, Kruss, Hamburg, Germany). The distilled water of the automatic distribution controller was dripped onto the sample to be tested (~5 μL), and 5 different positions were dripped onto each sample; each set of experiments was repeated five times. After 20 s (the contact angle decreases with time, and the optimum measurement time is 20 s), the Laplace-Young fitting algorithm was used to determine the mean value of WCA. Finally, the specimens were photographed and observed, and recorded the WCA of the image.

### 3.7. Optical Microscopy (OM) Characterization

The samples prepared above were observed by OM at room temperature using an inverted optical microscope (model: DMI3000B, Leica, Weztlar, Germany), and the voltage stabilizer was turned on 15–30 min in advance to preheat before using OM. The sample to be measured was placed on a clean slide, and the height of the stage was adjusted to a suitable position. The sample was first observed at 50×, and then the objective dial was adjusted to control the magnification of the sample, and finally a suitable objective was selected to observe the morphology of the sample.

### 3.8. Atomic Force Microscopy (AFM) Characterization

The surface morphology of the above prepared samples was characterized by AFM at room temperature, the roughness of the coating surface (Ra, arithmetic mean value chosen in this thesis) was determined by AFM (model: Multimode8, Bruker, Bremen, Germany), and the samples were characterized in Peak Force mode. The scanning range of AFM was set to 1 μm, and the scanning speed was set to 0.977 Hz. Five different regions were selected for a sample to preserve the data with uniform distribution of surface morphology changes, and the measured samples were processed using the NanoScope Analysis software.

### 3.9. Cell Adhesion Assay

The cell adhesion of the above prepared samples was determined by crystal violet staining at room temperature. Crystal violet, also known as gentian violet, is an excellent stain that stains the nucleus to show a blue color, the cytoplasm to show a pink color, and the whole cell to show a powder-blue color. Crystal violet has a wide range of applications in cytology, histology and bacteriology.

The pure Ti, Gelatin, G−SDS_6%_, G(SDS_6%_)−EPTAC, G(SDS_6%_)−DEQAS, G(SDS_6%_)−(MPA−N^+^), G(SDS_6%_)−DEQAS/(MPA−N^+^), G−STSo_6%_, G(STSo_6%_)−EPTAC, G(STSo_6%_)−DEQAS, G(STSo_6%_)−(MPA−N^+^), and G(STSo_6%_)−DEQAS/(MPA−N^+^) samples were placed in wells, and two additional sets of the same experiments were done (i.e., each experiment was repeated three times). Human umbilical vein endothelial cells (HUVECs, 5 × 10^5^ cells/mL) were inoculated in wells of cell culture plates, and the samples were placed at a temperature of 37 °C. The samples were incubated in RPMI 1640 medium with a temperature of 37 °C, a CO_2_ concentration of 5% and fetal bovine serum (FBS) of 10% at constant pressure for 24 h. The medium was then aspirated and the non-adherent cells were rinsed twice with PBS, while the adherent cells were fixed with 4% paraformaldehyde. Then, the cells were treated with 0.1% crystal violet staining for 5 min, followed by three washes with ddH_2_O. Finally, the cells were photographed and observed under an Axio Scope AI light microscope.

### 3.10. Cell Migration Rate Determination

Samples of pure Ti, Gelatin, G−STSo_6%_, G(STSo_6%_)−EPTAC, G(STSo_6%_)−DEQAS, G(STSo_6%_)−(MPA-N^+^), and G(STSo_6%_)−DEQAS/(MPA−N^+^) were placed in the wells of cell culture plates, and two other sets of the same experiments were performed (i.e., each experiment was repeated). HUVECs (2 × 10^5^ cells/well) were inoculated into the wells of the cell culture plates, and the plates were incubated in RPMI 1640 medium at a temperature of 37 °C, CO_2_ concentration of 5% and fetal bovine serum (FBS) of 10% at constant temperature and pressure for 24 h. The medium was aspirated with a sterile pipette, and Using a sterile pipette tip, make a vertical scratch right in the middle of the culture well, the cells within this scratch can be removed and then incubated in RPMI 1640 medium at 37 °C and CO_2_ concentration of 5% for 24 h. The number of cells in the scratch area was then observed under an Axio Scope.AI light microscope to determine the migration ability of cells.

### 3.11. Determination of Antimicrobial Resistance

*Escherichia coli* (*E. coli*) and *Staphylococcus aureus* (*S. aureus*) were grown to the mid-logarithmic stage. The bacterial suspensions were diluted to a concentration of 10^6^ CFU/mL. Samples of pure Ti, G(STSo_6%_)−EPTAC, G(STSo_6%_)−DEQAS, G(STSo_6%_)−(MPA−N^+^), G(STSo_6%_)−DEQAS/(MPA−N^+^) were placed in 5 mL of bacterial (*E. coli*, *S. aureus*) suspension and incubated at a constant temperature of 37 °C for 24 h. After incubation, the various samples were rinsed twice with PBS. The samples were soaked in 5 mL of PBS for 5 min, then 3 mL of the soaked bacterial solution was centrifuged in a centrifuge tube for 1 min with a pipette, and the lower layer was spread evenly on Mueller-Hinton agar medium for 12 h. Colony counting was performed.

## 4. Conclusions

Considering the limitations of Ti application in medical and industrial fields, gelatin-quaternary ammonium salt coatings were assembled on the surface of Ti to improve the surface biocompatibility and antibacterial properties of Ti and promote the bone integration ability of Ti implants. The structures of DEQAS and MPA−N^+^ were characterized by ^1^H NMR, and the grafting rates of quaternary ammonium salt were calculated by molar grafting rate. The aggregation morphology of polypeptide on a titanium surface was characterized by water contact angle (WCA) and atomic force microscope (AFM). The resulting coated titanium implant showed remarkable antibacterial functions, as well as significant promotions in cell adhesion and cell migration. The material has a broad application prospect in orthopedics, transplantation and surgery.

## Figures and Tables

**Figure 1 molecules-28-04570-f001:**
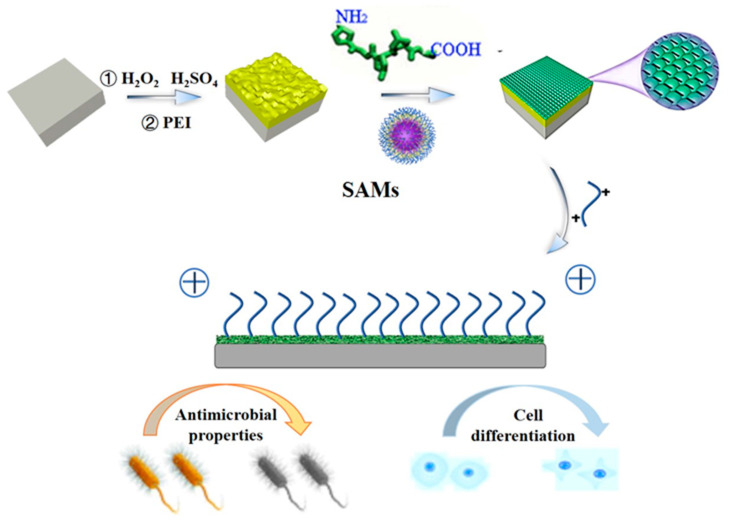
Mechanism Diagram of gelatin-quaternary ammonium salt coating prepared by self-assembly technique.

**Figure 2 molecules-28-04570-f002:**
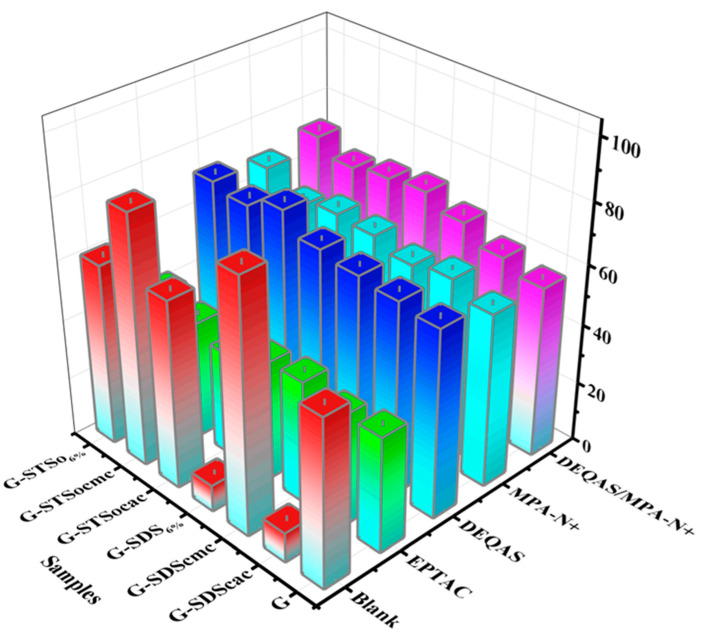
WCA of Gelatin Monolayers Grafted with Different Quaternary Ammonium Salts, *n* = 5.

**Figure 3 molecules-28-04570-f003:**
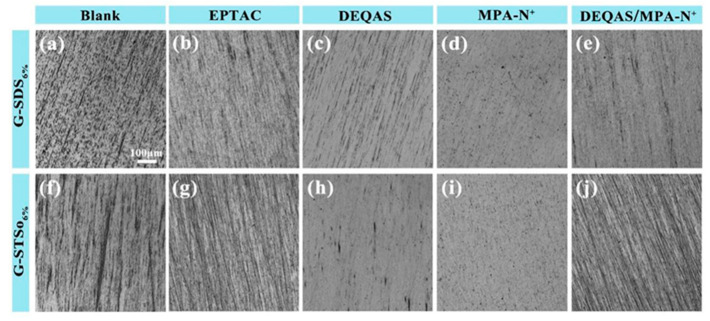
OM Image of Gelatin Monolayer Grafted with Different Quaternary Ammonium Salts ((**a**), G−SDS_6%_ gelatin monolayers, (**b**), G(SDS_6%_)−EPTAC, (**c**), G(SDS_6%_)−DEQAS, (**d**), G(SDS_6%_)−MPA-N^+^, (**e**), G(SDS_6%_)−DEQAS/MPA-N^+^, (**f**), G−STSo_6%_ gelatin monolayers, (**g**), G(STSo_6%_)−EPTAC, (**h**), G(STSo_6%_)−DEQAS, (**i**), G(STSo_6%_) MPA−N^+^, (**j**), G(STSo_6%_)−DEQAS/MPA−N^+^).

**Figure 4 molecules-28-04570-f004:**
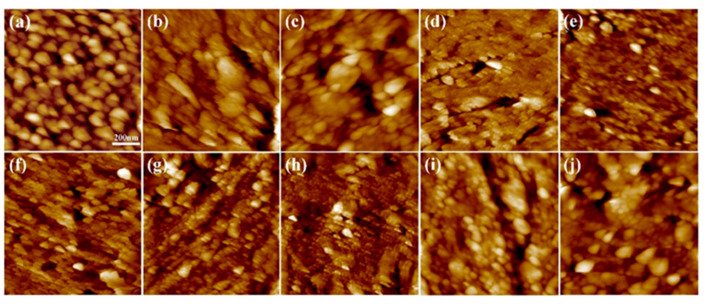
Surface Morphology Map of the Gelatin Monolayers Grafted with Different Quaternary Ammonium Salts ((**a**), G-SDS_6%_ gelatin monolayers, (**b**), G(SDS_6%_)−EPTAC, (**c**), G(SDS_6%_)−DEQAS, (**d**), G(SDS_6%_)−MPA-N^+^, (**e**), G(SDS_6%_)−DEQAS/MPA-N^+^, (**f**), G−STSo_6%_ gelatin monolayers, (**g**), G(STSo_6%_)−EPTAC, (**h**), G(STSo_6%_)−DEQAS, (**i**), G(STSo_6%_) MPA−N^+^, (**j**), G(STSo_6%_)−DEQAS/MPA-N^+^).

**Figure 5 molecules-28-04570-f005:**
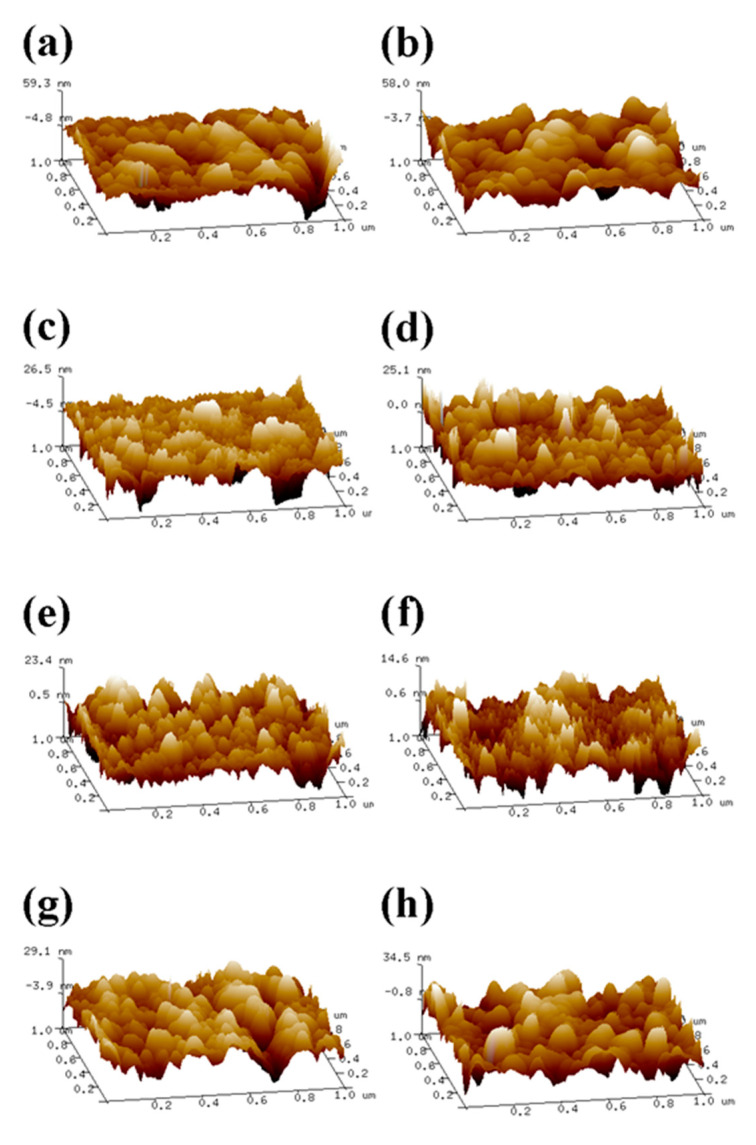
Surface 3D Morphology Map of the Gelatin Monolayers Grafted with Different Quaternary Ammonium Salts ((**a**), G(SDS_6%_)−EPTAC, (**b**), G(SDS_6%_)−DEQAS, (**c**), G(SDS_6%_)−MPA-N^+^, (**d**), G(SDS_6%_)−DEQAS/MPA-N^+^, (**e**), G(STSo_6%_)−EPTAC, (**f**), G(STSo_6%_)−DEQAS, (**g**), G(STSo_6%_)−MPA-N^+^, (**h**), G(STSo_6%_)−DEQAS/MPA-N^+^).

**Figure 6 molecules-28-04570-f006:**
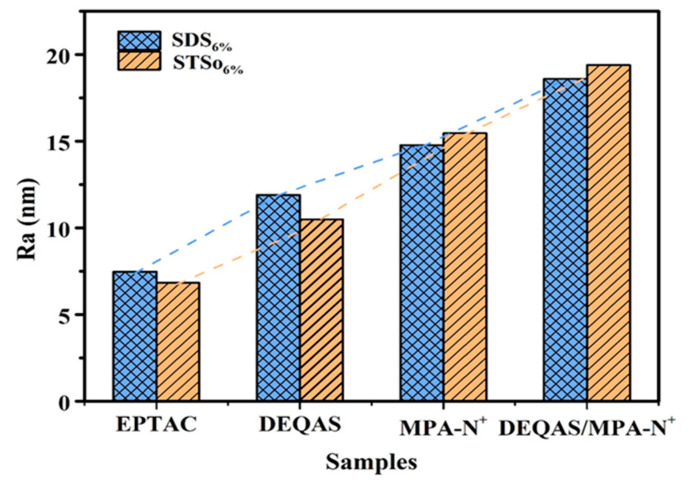
Ra of the Coating Surfaces of Gelatin Monolayers Grafted with Different Quaternary Ammonium Salts.

**Figure 7 molecules-28-04570-f007:**
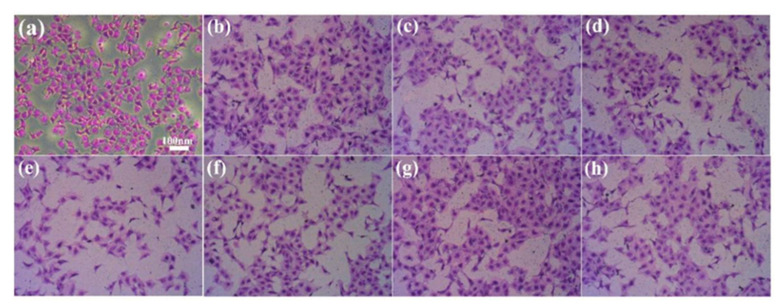
OM Images of Cell Adhesion on Different Coating Surfaces ((**a**), Control, (**b**), Blank, (**c**), Gelatin, (**d**), G−STSo_6%_, (**e**), G(STSo_6%_)−EPTAC, (**f**), G(STSo_6%_)−DEQAS, (**g**), G(STSo_6%_)−MPA−N^+^, (**h**), G(STSo_6%_)−DEQAS/MPA−N^+^).

**Figure 8 molecules-28-04570-f008:**
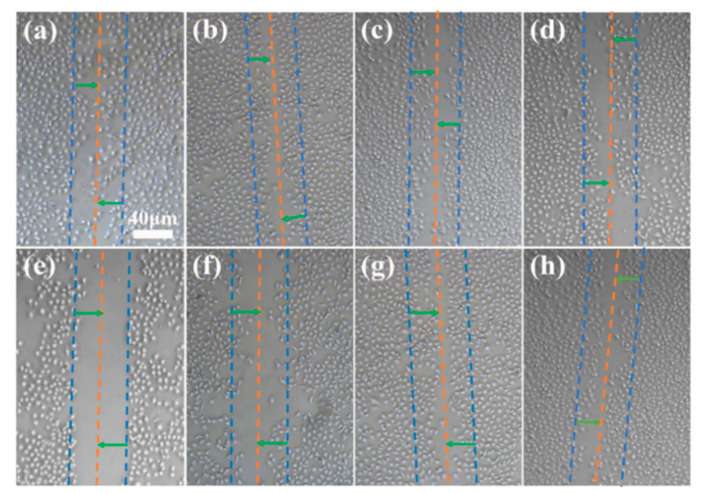
OM Images of Cell Migration on Different Coating Surfaces ((**a**), Control, (**b**), Blank, (**c**), Gelatin, (**d**), G−STSo_6%_, (**e**), G(STSo_6%_)−EPTAC, (**f**), G(STSo_6%_)−DEQAS, (**g**), G(STSo_6%_)−MPA−N^+^, (**h**), G(STSo_6%_)−DEQAS/MPA−N^+^).

**Figure 9 molecules-28-04570-f009:**
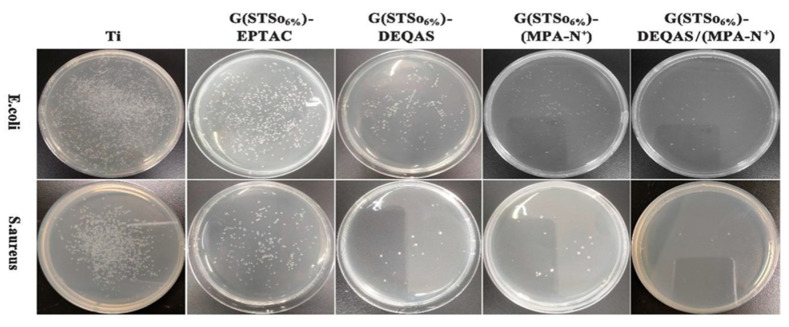
Colonies Map of *E. coli* and *S. aureus* on Different Coating Surface Samples after 24 h Culture.

**Figure 10 molecules-28-04570-f010:**
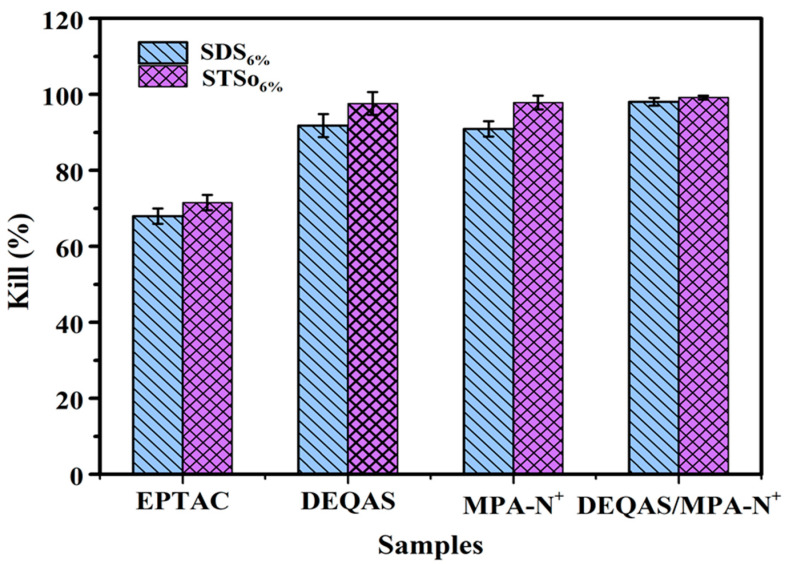
Bacteriostatic Rate Against *E. coli* and *S. aureus* on Different Coating Surface Samples after 24 h Culture.

**Figure 11 molecules-28-04570-f011:**
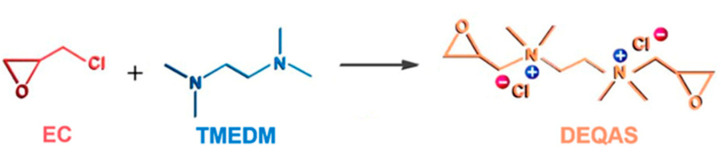
Synthetic reaction roadmap of DEQAS.

**Figure 12 molecules-28-04570-f012:**
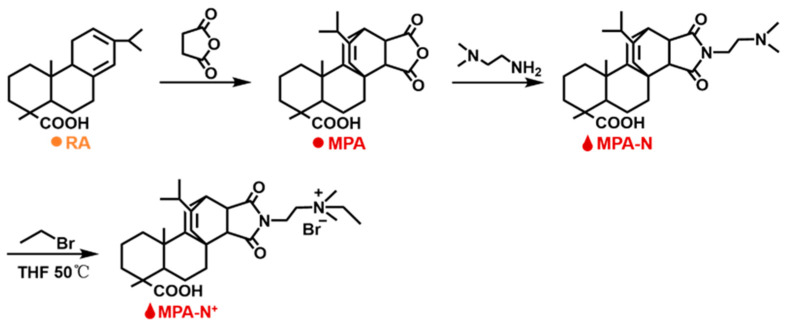
Synthesis reaction roadmap of MPA-N^+^.

**Figure 13 molecules-28-04570-f013:**
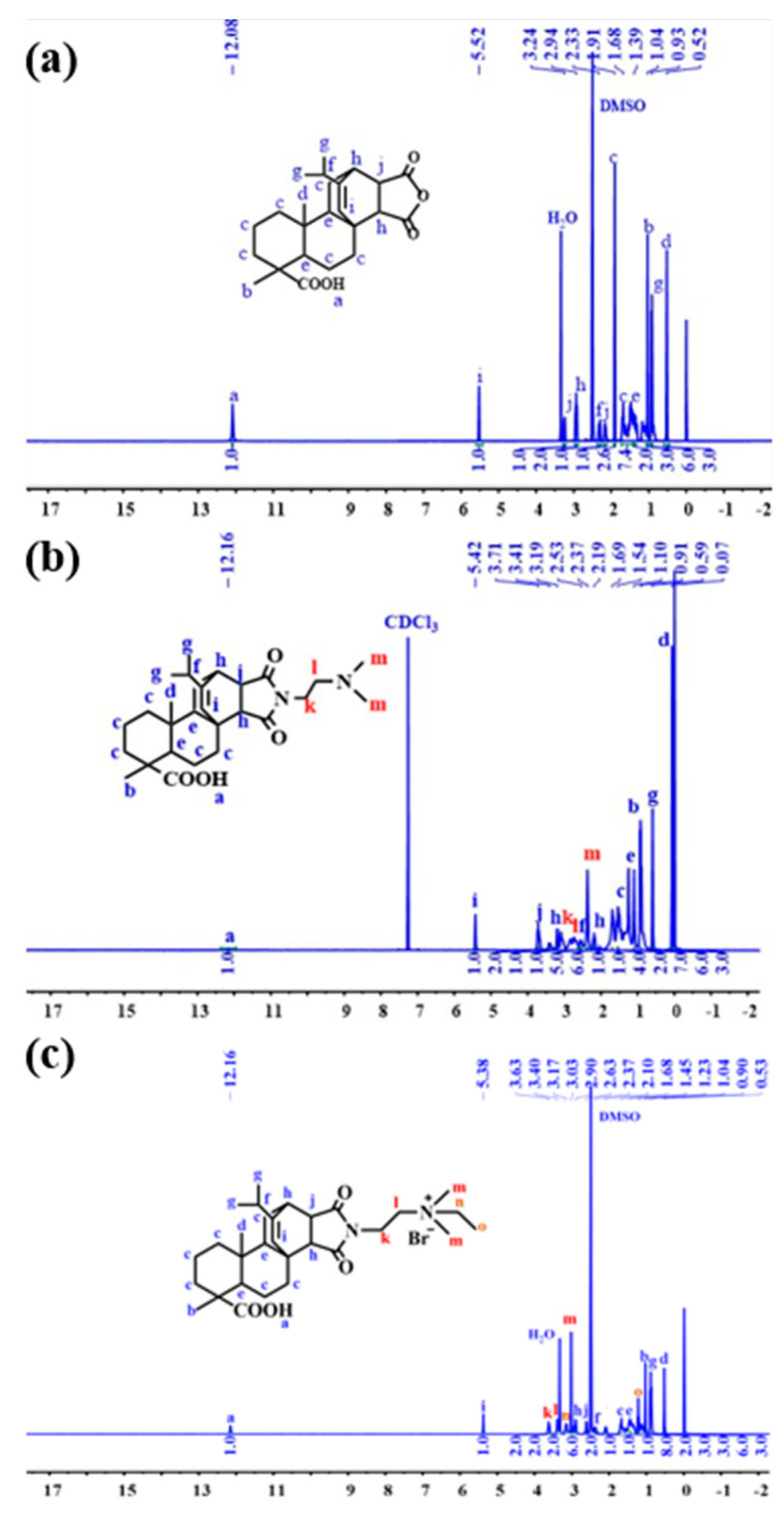
^1^H NMR Spectra of MPa, MPA−N and MPA−N^+^ ((**a**), MPA, (**b**), MPA−N, (**c**), MPA−N^+^).

**Table 1 molecules-28-04570-t001:** Molar Grafting Rate of Different Quaternary Ammonium Salts under Different Conditions (%).

	G						
	Blank	SDScac	SDScmc	SDS6%	STSocac	STSocmc	STSo6%
EPTAC	3.574	3.672	3.589	5.385	5.284	5.093	7.356
DEQAS	3.034	6.206	5.628	6.848	6.279	6.168	7.256
MPA-N^+^	3.627	5.482	5.583	5.632	5.965	6.065	6.353
DEQAS/MPA-N^+^	3.145	5.629	5.868	6.353	6.846	6.735	7.325

## Data Availability

Not applicable.

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
