# Peer review of "Preparation of Gelatin-Quaternary Ammonium Salt Coating on Titanium Surface for Antibacterial/Osteogenic Properties"

_molecules, 2023, doi:10.3390/molecules28124570_

Round 1

Reviewer 1 Report

General concerns

- Please carefully revise the English language in the manuscript to improve its accuracy and clarity for better readability.

- References in the manuscript must be thoroughly revised looking for missing references and non-proper numbering and citation in the text.

- Acronyms are usually introduced without any previous definition. This must be corrected.

Results and Discussion

Section 2.1. It is recommended to be included in Material and Methods.

-Line 107. References should be included here.

- What NMR signals distinguish MPA-N+ from MPA-N and MPA? This could be indicated for better clarity. Would be N-NMR spectra acquisition more conclusive than H-NMR, wouldn´t be?

- Figure 2. Please unify the peaks labelling for a more direct and clearer comparison between spectra (a)-(c). Redundant information in the figure´s footnote must be avoided.

 Section 2.2.

- Lines 119-120. “..was calculated as per the formula”.

Authors should clarify this or include any supporting reference.

- Lines 126-128. “…measurements showed that the exposure of -NH2 on the collagen polypeptide monolayer was positively correlated with…”.

Authors are encouraged to provide additional evidences that ammonium salts are covalently attached to the layers according to the reactions proposed, by using some spectroscopic characterization technique. QCM provides information about net mass changes that are related to any physisorbed and/or chemisorbed species as well as the exchange of solvent molecules and ions in the layers.

How can authors assure that grafting rates correspond only to covalently attached ammonium salts? This should be properly addressed and clarified.

 Section 2.3.

- What is the relationship between molecular weight (MW) and contact angle (CA)? Why increasing MW should also lead to the increase of CA?. Changes in CA usually relies on chemical interactions of the functional groups in the layer structure with water.

- Any reason for the anomalously low CA values for the G-SDS (6%) and G-SDScac compared to the rest of the samples? How many samples were prepared and tested by CA measurements per batch to check sample-to-sample reproducibility?

 Section 2.5.

- Different approaches and parameters are available to analyze surface roughness by AFM and to extract topographic information. Please justify, why the Ra parameter is only selected and the information that provides about coatings.

Materials and Methods

Section 3.1.

- This section is incomplete. Please carefully revise the chemicals missed including acronyms and their purity.

Section 3.2.

- What the preparation consists on the blank sample depicted in Table 1 and Figures 3 and 4?

 Section 3.3.

- This section should be revised to improve its readability and better comprehension to help the reader easily reproduce the experiments. The procedures and protocols must be simplified but better clearly described and detailed by avoiding unnecessary repetition of sentences and words.

- Lines 325. What is the PEI solution?

 Section 3.5.

- IR spectroscopy characterization is included but no IR spectra appear in this work. Were the coatings characterized by IR spectroscopy? If so, this should be described and included in the manuscript (Please check the comments in Section 2.2.)

- Line 391. What synthesized products were characterized by IR?

 Section 3.6.

- Some information is missed. For example: QCM materials and substrate characteristics, surface area, frequency, calibration constant, etc. Please include all the necessary information.

 Section 3.7.

- What is the initial drop volume and the evolution with time of the CA?

Please, justify the criteria for the CA values finally selected.

 Section 3.9.

- A more detailed description of the AFM methodology employed for the image analysis is suggested (e.g. number of different locations imaged per sample, statistical analysis, etc.).

 Conclusions

- This section should be revisited. In general, the conclusions are sometimes rather descriptive. Authors are encouraged to be more concrete and conclusive in the most significant results.

Please carefully revise the English language.

Reviewer 2 Report

This work has shown the novel surface engineering of quaternized ammonium on collagen polypeptide for improved osseointegration and biocompatibility of traditional Ti surfaces. The experimental method is scientifically sound and technique such as contact angle, AFM, cell adhesion, and plate colony counting was used to examine the material and antibacterial properties of the synthesized films. This work should be of interest to the biomaterial and medical community. The publication is recommended after some revision and proofreading to improve the quality of the paper. 

1. Please elaborate more on the novelty and significance of this work in the introduction section compared to other published literatures.

2. Section 2.7 and 2.8: Any data showing the cell migration and kill percentage of quaternary ammonium sample without any gelatin? If so, please elaborate on this.

3. Line 172: As shown in Fig 7 (this should be Fig 5?)

4. Some sentences are missing a period in between them. Please proofread the whole paper again. 

Please again proofread the whole paper for any grammar and punctuation errors. Some sentences are missing a period in between them.

Round 2

Reviewer 1 Report

General concerns

- Please carefully revise the English language in the manuscript, still to improve its accuracy and clarity.

- References in the manuscript must be still thoroughly revised looking for missing references and citation errors in the text. For example, cites 24 and 25 do not match with the comments in the text. There are several more similar examples.

Section 3.2.

-Line 309. Authors forgot to include any reference in place.

 Section 2.1.

-What the preparation consists on the blank sample depicted in Table 1 and Figures 2 and 3?

Response Without any treatment Ti is blank experiment.

Why should be bare Ti substrate used as blank for depositing the different ammonium salts? In figure 2, the letters G appears in the left axis assigned to the blank row. This fact, it seems to refer to the blank samples as Ti but also with a deposited gelatin layer on top, doesn´t it?. This reviewer is still confused about this. Please clarify this in the manuscript to avoid any misunderstanding of the readers.

 - Lines 114. “…was calculated as per the formula in parter 3.6”.

Instead of parter 3.6. should be section 3.5.

The reference 41 included by the authors does not deal with this matter. It should be either changed or eliminated.

 - Lines 120-122. “…measurements showed that the exposure of -NH2 on the collagen polypeptide monolayer was positively correlated with…”.

Authors cited references 41 and 42 to support the assertion made about the grafting of the ammonium compounds to the -NH2 groups exposed in the layers. Such references do not address properly such question. They should be either changed or eliminated. Instead, authors should only include the reference [44] (Journal of Hazardous Materials 383 (2020) 121142) and make some comments regarding their previous results obtained by IR spectroscopy at this respect.

 Section 2.2.

- Lines 132-135.

-What is the relationship between molecular weight (MW) and contact angle (CA)? Why increasing MW should also lead to the increase of CA?.

The author responds 1: Water barrier property and tensile strength were improved along with increased MW. References should be given in reference 43.

This is not correct, because the change in CA usually relies on chemical interactions with water of the functional groups in the layer structure. The reference [43] included by the authors do not support such assertion. It is recommended to revise or eliminate this part.

 -Any reason for the anomalously low CA values for the G-SDS (6%) and G-SDScac compared to the rest of the samples?

Response 2. The low CA values of G-SDS and G-SDScac (6%) may be due to the following reasons Water barrier property and surface roughness. Set the drip control at the computer end to 5 uL and drop 5 different spots on each sample.

Is this effect reproducible when preparing different samples of G-SDS (6%) or G-SDScac for measuring WCA? How many samples are measured in WCA for each type of layer? Authors missed this information.

Conclusions

- Line 504. Please delete IR.

Please carefully revise the English language in the manuscript, still to improve its accuracy and clarity.

Reviewer 2 Report

The authors have addressed previous comments in Report 1. However, there are two aspects of the paper that can be improved prior to publication. 

1. The authors must provide full spelling and definition of any abbreviation used in the main text when the terms appear for the first time, for example, OM, WCA, STSocac, STSo6%.

2. Please proofread again and correct any typo and grammar issues, for example, line 47 "ti".

Need to proofread for typos and grammar again. 
